# Effect of the Surface Chemical Composition on the Corrosion Resistance in the Mixture of FeCrMoNbB (140MXC) and FeCMnSi (530AS) Coatings Produced with the Electric Wire Arc Spraying Technique: Part I

**DOI:** 10.3390/ma16114182

**Published:** 2023-06-04

**Authors:** Héctor F. Rojas-Molano, Jhon J. Olaya-Flórez, María A. Guzmán-Pardo, José E. Alfonso-Orjuela, Néstor E. Mendieta-Reyes

**Affiliations:** 1Departamento de Ingeniería Mecánica, Facultad de Ingeniería, Universidad Libre–Seccional Bogotá, Bogotá 111071, Colombia; 2Departamento de Ingeniería Mecánica y Mecacatrónica, Facultad de Ingeniería, Universidad Nacional de Colombia, Bogotá 111321, Colombiamaguzmanp@unal.edu.co (M.A.G.-P.); 3Departamento Física, Facultad de Ciencias, Universidad Nacional de Colombia, Bogotá 111321, Colombia; 4Departamento de Química, Facultad de Ciencias, Universidad Nacional de Colombia, Bogotá 111321, Colombia

**Keywords:** coatings, corrosion, projection, FeCrMoNbB/FeCMnSi, 140MXC, 530AS

## Abstract

In this study, FeCrMoNbB (140MXC) and FeCMnSi (530AS) coatings were simultaneously projected on the substrate AISI-SAE 4340 using the electric wire arc spraying technique. The projection parameters, such as current (I), voltage (V), primary air pressure (1st), and secondary air pressure (2nd), were determined using the experimental model Taguchi L9 (3^4−2^). Its main purpose is to produce dissimilar coatings and to evaluate the effect of the surface chemical composition on the corrosion resistance in the mixture of 140MXC-530AS as commercial coatings. Three phases were considered to obtain and characterize the coatings: Phase 1: Preparation of materials and projection equipment; Phase 2: Coatings production; and Phase 3: Coatings characterization. The characterization of the dissimilar coatings was carried out using the techniques of Scanning Electron Microscopy (SEM), Energy Dispersive Spectroscopy (EDX), Auger Electronic Spectroscopy (AES), X-ray photoelectron spectroscopy (XPS), and X-ray diffraction (XRD). The results of this characterization corroborated the electrochemical behavior of the coatings. The presence of B was determined with the XPS characterization technique in the mixtures of the coatings in the form of Iron Boride. Moreover, the XRD technique showed Nb in the form of FeNb as a precursor compound for the 140MXC wire powder. The most relevant contributions are the pressures, provided that the quantity of oxides in the coatings decreases with respect to the reaction time between the molten particles and the atmosphere of the projection hood; moreover, for the corrosion potential, the operating voltage of the equipment does not exert any effect since these tend to remain constant.

## 1. Introduction

The phenomenon of corrosion is a problem that affects metal materials, especially ferrous alloys. The economic losses due to this problem reach 4% of GDP (gross domestic product) in the case of industrialized countries such as the United States [1] and 8% and 4% in Peru and Colombia, respectively [2]. It is well known that, since the 1970s, coatings have been the solution to these problems mentioned, with their greatest development in the 80s and 90s. Coatings have reached such a high technological level that applications are developed at the nanometric level [3]. This corrosive phenomenon can be minimized with thermal projection processes, which involve the contribution of deposited materials in the form of finely divided molten particles onto a properly prepared substrate [4]. In the electric wire arc spraying technique, the fusion speed and voltage determine the size distribution of the atomized particles, which is represented in the final microstructure of the coating [5]. The jet of compressed air projects the liquid metal as molten droplets onto the substrate [6], providing protective coatings against wear and corrosion [7], all of which are functions of its deposit parameters such as current, voltage, and primary and secondary air pressures [8].

The coating obtained by this technique is a superficial region of a material with properties different from those of the base material [9]. Its purpose is to replace, modify, lubricate, and optimize the surface of the material, giving it mechanical resistance to wear and corrosion [10]. A great advantage of the thermal projection by electric wire arc spraying technique is the wide variety of materials that can be used to make a coating. As a disadvantage, the area that the torch can cover is limited, resulting in size limitations as well as small and deep cavities into which no devices or extensions of the equipment fit [11].

Investigations such as [12,13,14,15] expose the influence of the projection parameters of commercial coatings deposited on low-carbon steel substrates, obtaining greater resistance against corrosion and wear. The coatings obtained with this technique were of the type monolayers, bilayers, and multilayers, always seeking to optimize the deposit parameters without changing the wires. The problem arose when studies began, where the combination between different commercial wires was sought deposited simultaneously to produce dissimilar coatings using the experimental design model “Taguchi”.

In the TWAS (Twin Wire Arc Spray) process, it is common to use compressed air as the atomizing gas. This air generally generates porosity, high surface roughness, and lower corrosion resistance [16]. When functional coatings with powder-cored wires are used combined with other coatings to spray simultaneously, the new composite coating not only has a good anti-wear effect but also maintains good corrosion resistance [17]. In the arc spray process (ASP), the heat source is the arc electricity obtained from the contact of two consumable metallic wires with different electric potentials, and the carrier gas is the compressed air. The velocity, dimensions, and thermal characteristics of the droplets sprayed are related to the morphology and properties of the coating [18].

COTECMAR (Science and Technology Corporation for the Development of the Naval Maritime and Fluvial Industry), as an institution representing the Colombian naval industry, installed TWAS equipment (SYSTEM EUTECTIC 4^®^) at the Bocagrande plant located in the city of Cartagena to spray commercial coatings such as 530AS^®^, 560AS^®^, and 140MCX^®^. Its purpose is to combine this type of coating to increase the useful life of its naval components in terms of resistance to corrosion and wear, as is often performed in offshore installations with organic coatings, which provide sufficient protection against corrosion and are well-established for applications in marine environments, including post-treatments that increase corrosion resistance [19].

## 2. Materials and Equipment

### 2.1. Substrate and Coatings

The materials used in this work were the following: (1) AISI/SAE 4340 (0.30–0.38% C, 0.40% Si, 0.50–0.80% Mn, 0.035% P, 0.035% S, 1.30–1.70% Cr, 0.15–0.30% Mo and 1.30–1.70% Ni) [20]; (2) the 530AS wire (0.21% C, 0.36% Si, 0.96% Mn, 87.78% Fe and 10.69% Cu) [21]; and (3) the 140MXC wire (0.44% C, 0.97% Si, 18.02% Cr, 1.08% Mn and 79.49% Fe) [22].

### 2.2. Thermal Spray Equipment

The equipment used for spraying coatings was an electric wire arc spraying device designed by Castolin Eutectic for cold-projected anticorrosion and anti-abrasion coatings with the designation *Eutronic Arc Spray 4*. Figure 1a shows TWAS equipment located in the COTECMAR facilities, and Figure 1b shows the schematic diagram of the equipment. This is a push-pull type action in which the wire feed unit is conveniently mounted on the primary power source; the drag of these is performed through a synchronizer installed in the projection gun with a flexible actuation position that feeds the reversible unit up to a distance of 20 m. Its operating functions, such as the projection intensity, voltage, speed of advance, and air pressures, are parameterized by the console located at the front of it.

### 2.3. The Equipment Used for the Corrosion Tests and Their Characterization were

#### 2.3.1. Saline Chamber Equipment

Saline chamber tests were performed using the CCT DIES equipment with an exposure time of 1200 h and a saline solution of NaCl. (The salt solution was prepared by dissolving 5 ± 1 parts by mass of sodium chloride in 95 parts of water conforming to Type IV water in ASTM B 117: Panreac reagent with 99% purity, distilled and deionized water with a conductivity of 0.67 μS/cm, pH of the solution 6.8 at 21 °C, working temperature 35 ± 1 °C, pressure 10 to 20 psi, air flow 7 ft3/min, and solution flow at 0.5 L/h.)

#### 2.3.2. Potentiodynamic Polarization Equipment

A potentiodynamic polarization (Tafel) test was performed with GAMRY Instruments 600 Potentiostat/Galvanostat equipment, the volume of the electrochemical cell at 100 mL, NaCl solution at 3%, room temperature 20 ± 1 °C, working area of 0.196 cm^2^, Calomel reference electrode, platinum bar auxiliary electrode, the potential between −0.5 V and 0.7 V at a rate of 0.5 mV/s, and stabilization time in 60 min.

#### 2.3.3. Electrochemical Impedance Spectroscopy Equipment

An electrochemical impedance spectroscopy (EIS) test was performed with the same TAFEL test parameters plus a voltage of 10 mV AC; frequency between 0.01 Hz and 1 MHz; and times of 24, 48 and 168 h to verify if there were significant differences in the impedance by a possible misapplication of sample collection.

#### 2.3.4. Scanning Electron Microscope Equipment

The chemical composition of the base coatings and mixtures was established with the FEI-QUANTA 200 equipment. Operation variables included: high vacuum, voltage of 30 Kv, pressure of 3 × 10^−7^ torr, and a magnification of 200−20 µm.

#### 2.3.5. Auger Electron Spectroscopy Equipment

The surface chemical composition of the coating mixtures was carried out using the Auger electron spectroscopy (AES) technique. The equipment used was an EA-125-U7-Hemispherical Electron Spectrometer Omicron Nano Tech AES-LD\30028. As operating variables, the following were considered: start energy of 20 eV, end energy of 1500 eV, energy interval of 0.2 eV, residence time of 50 min, phase of 245, and an amplitude of 7.

#### 2.3.6. X-ray Photoelectron Spectroscopy Equipment

For the XPS technique, the Binding energy analysis was performed using a Kratos Axis/Ultra DLD device. Operating variables were: Al Kα radiation, voltage 400 to 1490 eV, pressure equivalent to 10-10 Torr, cleaning with Ar, and a current of 300 mA. In order to verify the presence of these elements and especially the one of Si, B, Nb, and Mo on the surface of the 140MXC-530AS mixtures, the technique of X-ray photoelectron spectrometry (XPS) was used, and the behavior of its chemical environment was determined, consulting the database http://srdata.nits.gov/xps/ (accessed on 16 March 2020). The basic identification of the elements in the mixtures made with the CasaXPS^©^ program. Internal analysis was performed of the energy levels present in the peaks of the mixtures was made with the XPSPEAK 4.1^©^ program, adjusting the signals (Gaussian 70%—Lorentzian 30%) to reach more exact values corresponding to their binding energy [23].

#### 2.3.7. X-ray Diffraction Equipment

The X-ray diffraction (XRD) test was performed with Bruker AXS-D8 Advanced X-Pertpro Panalitycal equipment. The operating variables were as follows: Ɵ–2Ɵ configuration, range from 10° to 120°, voltage of 45 kV, current of 40 mA, monochromatic radiation Cu Kα, a wavelength of 1.56 Å, and step of 0.02°. The microstructure, present phases, and preferential crystalline orientations were established for the base coatings and for the 140MXC-530AS mixtures to corroborate the information obtained in the chemical identification phase; the indexation of the diffraction patterns was performed according to the JCPDS-ICDD (Joint Committee on Powder Diffraction Standards and International Center for Diffraction Data) letters. 

#### 2.3.8. Evaluation of Corrosive Conditions Performed According to Designation B117–19 Standard Practice for Operating Salt Spray (Fog) Apparatus

The procedure involves the exposure of steel test panels and the determination of their mass losses in a specified period of time. The required test panels, 76 mm by 127 mm by 0.8 mm (3.0 in. by 5.0 in. by 0.0315 in.), are made from commercial-grade cold-rolled carbon steel. Preparation of panels before testing: Clean panels before testing by degreasing only so that the surfaces are free of dirt, oil, or other foreign matter that could influence the test results. After cleaning, weigh each panel on an analytical balance to the nearest 1.0 mg and record the mass. 

Positioning of test panels: Place a minimum of two weighed panels in the cabinet, with the 127 mm (5.0 in) length supported 30° from vertical. Place the panels in the proximity of the condensate collectors. Duration of the test: Expose panels to the salt fog for 48 h to 168 h. Cleaning of test panels after exposure: After removal of the panels from the cabinet, rinse each panel immediately with running tap water to remove salt, and rinse in reagent-grade water. Chemically clean each panel for 10 min at 20 °C to 25 °C in a fresh solution prepared as follows: Mix 1000 mL of hydrochloric acid with 1000 mL reagent-grade water and add 10 g of hexamethylene tetramine. After cleaning, rinse each panel with reagent-grade water and dry. Immediately after drying, determine the mass loss by reweighing and subtracting panel mass after exposure from its original mass.

## 3. Method

Figure 2 methodologically summarizes in a flow chart the activities carried out in the 3 main phases of the research.

As for the Taguchi methodology used in this work, we worked with a fractional factorial with an orthogonal array L9 (3^4−2^). The factors considered were the parameters of current projection: voltage, primary pressure, and secondary pressure. The levels were established in the low, medium, and high operating ranges for each of the parameters as follows: current at 120, 140, and 160 A; voltage at 28, 30, and 32 V; primary pressure at 3.4, 4.2, and 4.8 Bars; and secondary pressure at 3.4, 3.8, and 4.2 Bars. With these values, the Taguchi matrix gave a total of nine experiments with the combination of levels and factors.

## 4. Results and Discussion

### 4.1. Chemical Identification of the Coatings and the 140MXC-530AS Mixtures

The chemical identification of the coatings and mixtures thereof (140MXC-530AS) was performed with the EDX measurement probe SEM equipment; elements of the base coatings with the characteristic peaks of each and values are shown in Figure 3. In the case of the 530AS^®^ coating, which is a continuous solid, Cu peaks are evident, which are incorporated in the surface of the wire as a protective film against corrosion [24] (Figure 3a), while the 140MXC^®^ coating (Figure 3b) is a discontinuous solid (tubular cover), allowing the incorporation of alloying powders (nanocrystalline compounds SEM detail Figure 3c), which serve as nucleators of the crystals as reported by [25] in their investigations. Another characteristic from the morphological point of view is that these powders are completely irregular [26] and tend to remain agglomerated with each other due to the use of organic binders in them [27]. Figure 3d shows the EDX spectrum for all the experiments, as well as the most representative elements in this mixture, 140MXC-530AS.

In the case of 140MXC wire nanocrystalline powder, the chemical composition obtained by the EDX test was as follows: 3.16% C, 3.13% O, 2.05% Si, 18.86% Cr, 1.75% Mn, 64.95% Fe, 2.71% Nb, and 2.88% Mo. The chemical composition for the mixture of coatings also by EDX test was as follows: 2.23% C, 3.51% O, 1.15% Si, 11.21% Cr, 0.91% Mn, 78.87% Fe, 1.55% Nb, and 0.55% Mo.

The variation between the initial chemical composition of the 530AS and 140MXC coatings and the mixture of both allowed us to establish that the projection parameters of TWAS equipment had an influence on the mixtures obtained in each of its elements in the following way: The electric current caused C, Si, Mo, Cr, Mn, and Fe to decrease, while O and Nb increased. Voltage caused C, Si, Cr, and Fe to decrease and O, Nb, Mo, and Mn to increase. The increase in oxygen is explained by the condition of oxidation of the elements that tend to decrease when the fusion of the wires occurs, as manifested by [28]. With primary pressure, it was observed that C, O, Nb, Mo, and Mn increased; Fe decreased; Si and Cr remained constant. Secondary pressure Si and Mo increased; Mn and Fe decreased; and C, Nb, and Cr remained constant. It can be established that the atomization gas exerts a lesser influence on the elements after the electrical parameters (current and voltage), with some elements coming from the nanocrystalline powders of the 140MXC^®^ remaining stable during its permanence in the air flow of the projection hood at the time of application, as reported by [29].

In order to study the surface of the deposits, the Auger Electron Spectroscopy (AES) technique was used. Table 1 presents a summary of the average intensities relative energy (keV) for Lα and Kα levels of the elements present in the coating mixtures.

It can be seen that most of the elements of the base coatings are maintained, with the exception of the boron that did not reach detection. The Kα peaks of Nb and Mo at 16.5840 and 17.4446 KeV were also not reflected in these energy values, but they were seen in a very slight way in the Lα levels at 2.1659 and 2.2932 KeV, respectively, along with the Si (Kα) at 1.7398 KeV [30]. 

For this purpose, the X-ray photoelectron spectrometry (XPS) technique was also used, as seen in Figure 4. This was performed in order to verify the presence, especially of B, on the surface of the 140MXC-530AS coatings detected in the AUGER technique and to determine the chemical behavior of its environment in the nine experiments. All the experiments show the same behavior in the XPS spectrum, and the binding energy values for the identified elements were as follows: Fe (710 eV), Mn (640 eV), Cr (575 eV), O (530 eV), Mo (400 eV), C (284 eV), Nb (205 eV), Si (102 eV), and B (191 eV). 

The XPS technique confirms that the peaks of the elements Fe, Mn, Cr, Mo, and Si show energetic states 2p and 3p; O, C, and B show energetic states 1s; and Nb shows energetic state 3d. Si, Nb, B, and Mo have characteristic electronic energies of 92, 167, 179, and 186 eV, respectively, which allows them to display double energy states at each energy level due to spin/orbit doubling; this agrees with [31,32]. 

Regarding O/1s, it was considered for the adjustment of the possible mixed oxides present in the mixtures with a value of ±531.1 eV. The Si is presented with a 3p_3/2_ electronic structure in the form of SiC in ±101.5 eV; briefly, the B shows the 3s electronic structure in the form of FeB in ±91.8 eV because the Fe also has a 3s electronic structure in 92 eV.

The Nb exhibits the 3d_5/2_ electronic structure in the form of Nb_2_O_5_in ±206.7 and 207.0 eV and of Nb in ±202.4 eV, and Mo manifests a 3p_3/2_ electronic structure in the form of MoO_2_ in ±398.5 eV and of MoO_3_ in 399.5 and 396.2 eV, respectively. In the case of Fe, there is the presence of Fe_2_O_3_ with the 2p_1/2_ electronic structures in ±724.0 eV and 2p_3/2_ in ±710.4 eV. 

The Cr is presented with a 2p_3/2_ electronic structure in the form of Cr_2_O_3_ in ± 578.3 and 576.7 eV, while the very attenuated Mn performs it with a 2p_3/2_ electronic structure in the form of MnO_2_ in ±642.1 eV, according to [33].

The C/1s signal was taken as a reference for the calibration of the other readings; its exact value was 284.7 eV with an associated error of ±0.1 eV with reference to the value of 284.6 eV for the aliphatic carbon used [34].

### 4.2. Microstructural Characterization of Base Coatings and the 140MXC-530AS Mixtures

Figure 5 shows the diffraction patterns of the base coatings compared with the ones of the substrate. The 530AS^®^ wire has six peaks in the diffraction pattern; three of these correspond to Fe(α) with BCC crystalline structure, crystalline planes ((1 1 0), (2 0 0), (2 1 1)) located at 44.5855°, 64.8152° and 82.2359° (JCPDS-ICDD 00-006-0696 letter). The remaining three peaks concern the FeCu with the FCC crystalline structure, crystal planes ((1 1 1), (2 1 4), (2 2 0)) located at 43.2502°, 50.2889° and 74.0313° (JCPDS-ICDD 00-004-0836 letter); these parameters are similar to those exposed [35]. The 140MXC^®^ wire shows six peaks in the diffraction pattern, but with the crystalline planes located at 44.4646°, 64.7872°, and 82.0523°, respectively, and the other three due to the very incipient reflections, possibly because of the nanocrystalline compounds related to FeNb with BCC crystalline structure, crystalline planes ((1 0 2), (1 1 1), (2 1 4)) located at 37.6933°, 41.4852° and 50.6932° (JCPDS-ICDD 00-001-1203 letter); these parameters coincide with the research carried out by [35]. 

Figure 6 shows the diffraction patterns for the nine experiments, including the substrate; all the mixtures show the characteristic peaks of Fe(α) with BBC crystalline structure, crystalline planes ((1 1 0), (2 0 0), (2 1 1)) located with respect to 2θ in 44.3071° (of greater intensity) and in 64.8152° and 82.2359° (of smaller intensities) according to the JCPDS letters: ICDD 00-006-0696, 01-087-0721, and 00-001-1262 letters. The other identified peaks correspond to Fe2O3(α) (hematite) with a rhombohedral crystalline structure, crystalline planes ((0 1 2), (1 1 0), (1 1 6)) located at 24.9218°, 35.6126° and 49.5113° with very low intensity compared to those of Fe(α) coinciding with the JCPDS-ICDD 00-029-0713, 00-034-1266, and 00-033-0664 letters, as reported by [36].

The attenuation in the XRD signals of the hematite may be due to the relatively high cooling rate in the transport phase of the particles and the effect of the different sizes of the nanocomposite powders that serve as nucleators of the crystals influenced by the pressures of projection, as [37] states in his investigations. On the other hand, the projected particles support during the transport phase a microstructural transformation outside the atomization that, at the time of the impact with the substrate, is recrystallized [38].

This can be explained from the thermal point of view of the particles in the transport phase. Figure 7 presents an infrared thermal image taken from the electric arc spray process.

The isotherms show the highest temperature of the particles in the central part of the projection bell, possibly due to the concentration of particles that travel closely with respect to each other (reddish color). As the particles move away from the midline of the projection hood, the temperature decreases (colors from yellow to violet), which contributes to their surface cooling faster and increasing their solid fraction ratio, especially for those of smaller size, according to [39]. The highest temperature recorded for the particles (803 °C) was around the fusion zone of the wires and in the central part of the projection cone, which implies immediate oxidation on the surface of the particles due to thermal effects and generates highly exothermic reactions [40].

### 4.3. Corrosion Resistance of 140MXC-530A Mixtures

#### 4.3.1. Saline Chamber Test

With the saline chamber test, the mass loss of the coatings in the 140MXC-530AS mixtures was determined. Figure 8 shows the morphology of the mixtures exposed to the saline chamber, evidencing the initial superficial condition of the coating before the test with the formation of uniform splats (Figure 8a); after 1200 h of testing, the coating was coated with a layer of rust (Figure 8b), which was removed by ultrasonic cleaning, observing the degradation of the material (Figure 8c). Subsequently, the SEM analysis at 20 μm (Figure 8d) showed the affectation of the splats with the formation of microcracks (red arrows), the disintegration of the semi-molten particles (blue arrows), and the porosity magnification of the coating (yellow arrows). The variation in the mass loss was between 0.1051 g for experiment No 3 and 0.6334 g in experiment No 1; this procedure was carried out according to the mandatory annex of the ASTM B 117 standard, which states, “*Determining Mass Loss*—Immediately after drying, determine the mass loss by reweighing and subtracting panel mass after exposure from its original mass”. This condition is due to typical defectology as pores derivatives of the TWAS technique, which is where the degradation mechanism of the material starts, having an impact on the increase in these and in the generation of the microcracks, along with the deterioration of the semi-molten particles, especially for the chemical characteristics of the 530AS^®^ wire that facilitates the action of corrosion as reported by [41].

In this case, the mass loss is more influenced by the pressures, although the electrical parameters, especially the current, can be explained from the point of view of the thermal conductivity of the precursor coatings and have an inverse relationship with the increase in the oxide inclusions generated during the projection, so the minimum oxide contents cause the highest thermal conductivity in the coatings according to the research of [42]. From the point of view of the thermal conductivity of the coatings, the temperature tends to increase for large particles, i.e., greater than 200 microns. This depends on the atomizations that the particles can suffer according to the shear stresses reached in the wires when the drop is generated as molten material to be transported by the air flow in the projection hood, as reported by [43].

Another explanation from the chemical perspective of the mixtures is related to the oxidation potentials of the elements present in them, which translates into the loss of ions such as Fe → Fe^2+^ + 2 when coming into contact with the atmosphere of the saline chamber, causing the coating to behave as an anodic system or sacrificial coating resulting from the formation of an electromotive force that retards the affectation of the base material or substrate, according to [44]. For pressures, it is attributed to inhomogeneity in the particle size, which largely affects the final porosity of the coatings [45].

#### 4.3.2. Potentiodynamic Polarization (TAFEL) Test

The equilibrium or polarization corrosion potential (Vf) was determined in the TAFEL test with respect to a reference electrode that allowed for establishing the susceptibility of the coatings to corrode in the given electrolytic environment. Figure 9 shows the potentiodynamic scan for all nine experiments of these coatings, compared against the substrate. The potential of the mixtures is in the range of ≈−600 mV and ≈−400 mV, while the current density is between ≈0.8 µA and ≈25 µA. In the mixtures of the coatings, an increase in the resistance to corrosion is observed, which is reflected in the increase in the potential for corrosion and a decrease in the exchange current, which indicates a lower flow of electrons from the material under test. It is here where the formation of oxides is estimated, especially Cr_2_O_3_, as analyzed with the XPS technique, and whose formation is given by the Cr coming from the 140MXC wire [46]. 

In this case, experiments 3, 7, and 5 are favored over corrosion resistance by presenting the lowest corrosion current values. The corrosion rate of coatings as a function of exposure time is influenced by corrosion potentials as they become closer to zero (0). For this case, experiment 3 of the mixtures is more favored against the resistance to corrosion by presenting the values of current density lower (≈0.85 µA) and with a potential that comes closer to zero (≈425 mV), influencing in the corrosion rate of the coatings as a function of the exposure time. Table 2 shows the summary of the potentials and corrosion currents of the substrate and of the mixtures of the coatings. This behavior can be explained from the point of view of the velocity of flight of the particles since these influence the reduction in the corrosion current when the particles impact with the substrate generating splats with flatter extensions, increasing the adhesion between them, obtaining coatings with high cohesion between the splats, fewer pores and oxides, resistant to corrosion, and of high quality [47].

#### 4.3.3. Electrochemical Impedance Spectroscopy (EIS) Test

Electrochemical impedance spectroscopy EIS is an electrochemical method that allows the characterization of the properties of materials and electrochemical systems. This technique consists of applying a sinusoidal disturbance of variable frequency potential to the material studied, after which its current response is recorded in the form of another sinusoidal signal inside an electrolytic cell. Figure 10 shows the Bode diagram for experiment No 8, along with the impedance phase angle diagram of the high-frequency band of 140MXC-530AS mixtures.

These results can also be interpreted as a function of the logarithm of the impedance magnitude (left vertical axis), the phase angle (Φ) (right vertical axis), and the logarithm of the frequency (horizontal axis); it is evident that the immersion time of the samples in the electrolytic solution generates a resistance between this media and the working electrode (Rsol), having an influence on the resistance to polarization (Rcorr or Rp) and on the total impedance of the system for form pores (Rpor). The electrical polarization suffered by the coatings takes longer to take place due to the ion exchange of the elements present in them and their concentration at a superficial level. Inevitably, corrosion is not a process that occurs through simple and uncomplicated active dissolution, especially during prolonged exposure periods; reactive materials, such as Fe alloys, corrode in solutions with a pH between 4 and 9, especially when the oxide/hydroxide solubilities are low and tend to accumulate deposits of corrosion products according to [48]. This is due to the fact that the resistance to polarization increases as a function of time, corresponding to the displacement of the impedance; this phenomenon depends on the mechanism of localized corrosion and the level of defectology present in the deposits (pores, microcracks, etc.) as a product of the projection parameters according to what was reported by [49].

The equivalent circuit modeling the EIS test function Bode diagram is shown in Figure 11; this is the resistive/capacitive type, which conforms to the surface condition of these mixtures, and in Figure 12, the Nyquist diagram is adapted for this type of coatings. It is necessary to consider that the electrochemical impedance characteristic for the coating mixture is a function of the solution prepared at room temperature and that the Nyquist diagram exhibits a capacitive loop at frequencies between 100 KHz and 0.1 Hz, according to the Bode diagram shown above that represents the charge-controlled corrosion process and an inductive loop at low frequencies that is associated with the relaxation of corrosion products.

Figure 13 illustrates the corrosion mechanism present in the 140MXC-530AS mixtures; the relationship between the corrosion potential that tends to decrease as the current density increases allow pitting corrosion to manifest more easily, nucleating on the surface of the coating and propagating inwards through the microcracks generated by the volumetric contraction and the residual efforts that the splats undergo during their cooling, generating a mechanism of electrolyte dysfunction in the thickness of the coating (Figure 13a) according to [50]. It should also be considered that 140MXC-530AS mixtures are obtained from their precursor wires with different physicochemical and mechanical properties (Figure 13b); there is a high probability of generating galvanic-type corrosion, making this an indicator for the loss of passivity or breakage of the passive layer according to [51]. The average values of the corrosion profile made for this case were the following: C ± 1.87%, O ± 3.89%, Si ± 0.94%, Nb ± 1.15%, Mo ± 0.45%, Cr ± 9.21%, Mn ± 0.79%, Fe ± 73.63%, Cl ± 3.47%, Na ± 3.11%, and S ± 1.49%. In this case, the Cl and Na correspond to the residues of the saline solution used in the EIS test, and the S is taken as a contaminating element, possibly during the manipulation of the sample and the likely formation of FeS_2_ as a product of corrosion [52], as well as that one of the respective oxides of Si, Nb, Mo, Cr, Mn, and Fe [53].

## 5. Conclusions

From a chemical point of view, with the EDX technique, the presence of most of the elements in the mixtures was confirmed, except Boron, which was not seen in this technique. For this reason, it was necessary to use the AUGER technique that allowed its detection at 179 eV. Si, Nb, and Mo were also detected with characteristic electronic energies of 92, 167, and 186 eV, respectively, which allows them to display double energy states at each energy level due to spin/orbit doubling. Additionally, with the XPS technique, the behavior of the chemical environment of the coating was determined, showing the energetic states 2p and 3p for the elements Fe, Mn, Cr, Mo, and Si; 1s for the elements O, C, and B; and 3d for the Nb.

Based on the microstructural characterization, all the mixtures of the coatings presented the characteristic peaks of Fe(α) with a BCC crystalline structure. This causes the resistance to corrosion to be influenced by the sprayed particles, which are transformed microstructurally in the transport phase, where they are atomized in the spray hood by the gas pressures until they hit the substrate and undergo recrystallization.

The corrosion resistance of the coating mixtures was correlated with the chemical composition of the wires (base coatings). The corrosion potential decreased significantly thanks to the presence of elements such as Cu (10.69%) in the 530AS wire and Cr (18.02%) in the 140MXC wire. Additionally, elements such as Cr (18.86%) and Nb (2.71%) present in the nanocrystals of the powder in the 140MX wire make the mixtures of the coatings increase their resistance to corrosion. The results of polarization resistance measurement have shown that 140MXC-530A mixtures provide low corrosion rates, indicating excellent corrosion resistance; experiment 3 was particularly the most favored in terms of the TAFEL test.

Although it is evident that the mixtures of the coatings exhibited preset corrosion in the salt chamber, tafel, and EIS tests, the corrosion process resulted in a delay in the load transfer process, with a decrease in the corrosion rate on the coating layer, due to impedance and phase angle.

It is also necessary to consider that the resistance to corrosion was evaluated in a direction perpendicular to the joint of coating and substrate because this property is not isotropic due to the geometry and arrangement of the splats that form the coating, as represented by the corrosion mechanism shown in Figure 13. For this reason, it was possible to establish that isolated corrosion was generated at the limits of the splats because these are considered more energetically active. 

## Figures and Tables

**Figure 1 materials-16-04182-f001:**
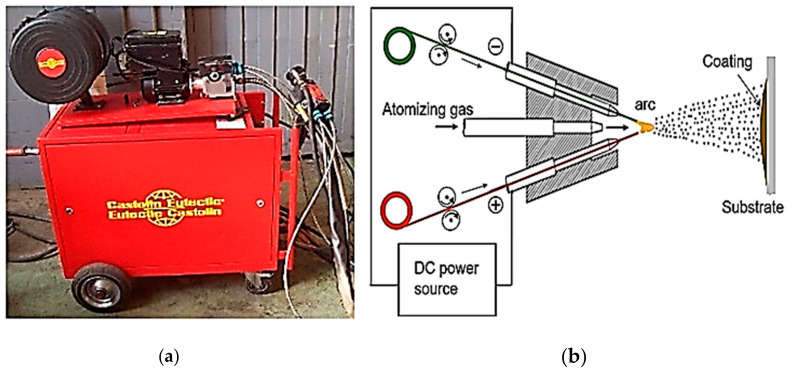
(**a**) TWAS equipment (Electric wire arc spraying equipment (with permission from COTECMAR)); (**b**) schematic diagram TWAS equipment (Schematic diagram of electric wire arc spraying equipment (Source: https://n9.cl/oicb3m, accessed on 18 May 2023)).

**Figure 2 materials-16-04182-f002:**
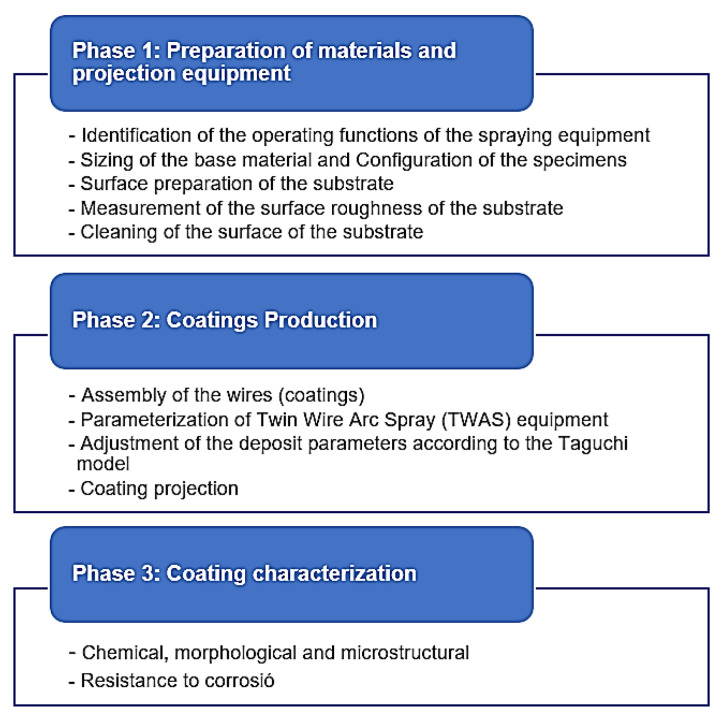
Flow chart of the research.

**Figure 3 materials-16-04182-f003:**
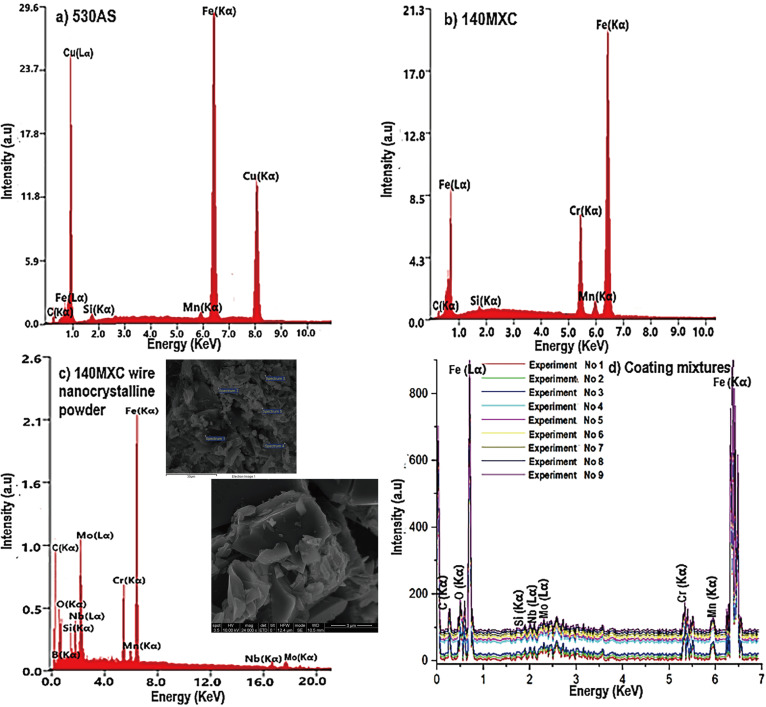
EDX spectrums for the base coatings and the 140MXC-530AS mixtures: (**a**) 530AS^®^ wire; (**b**) 140MXC^®^ wire; (**c**) 140MXC^®^ nanocrystalline powders; (**d**) 140MXC-530AS mixture.

**Figure 4 materials-16-04182-f004:**
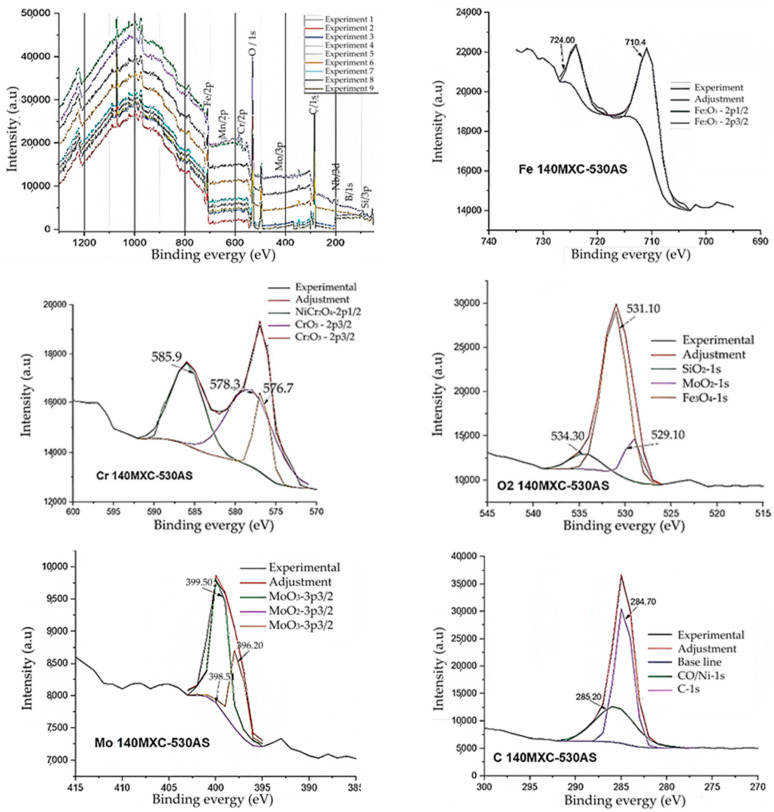
XPS spectrums for the 140MXC-530AS mixture.

**Figure 5 materials-16-04182-f005:**
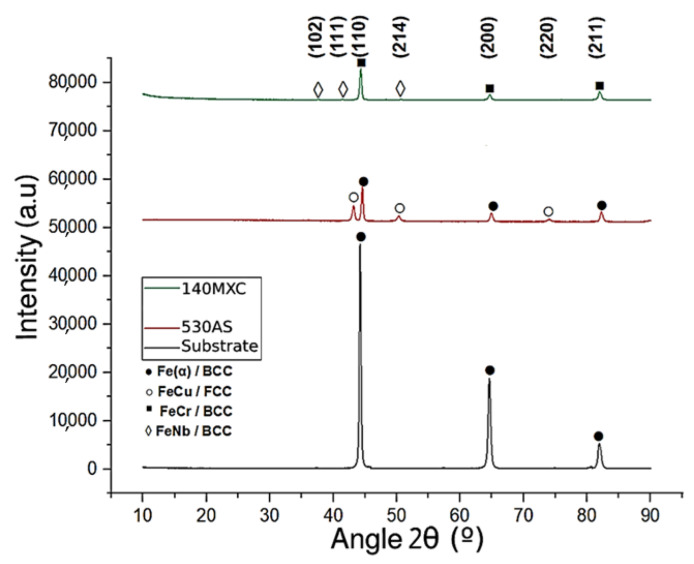
Diffraction patterns of the substrate and base coatings.

**Figure 6 materials-16-04182-f006:**
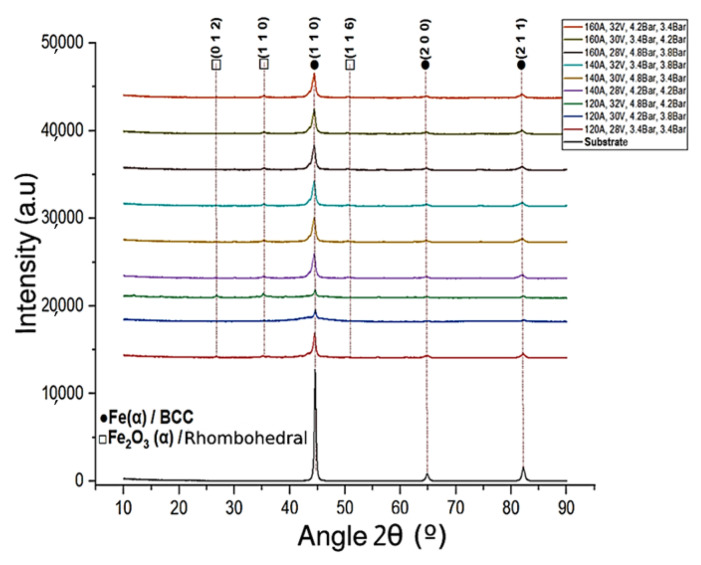
Diffraction patterns for the 140MXC-530AS mixtures.

**Figure 7 materials-16-04182-f007:**
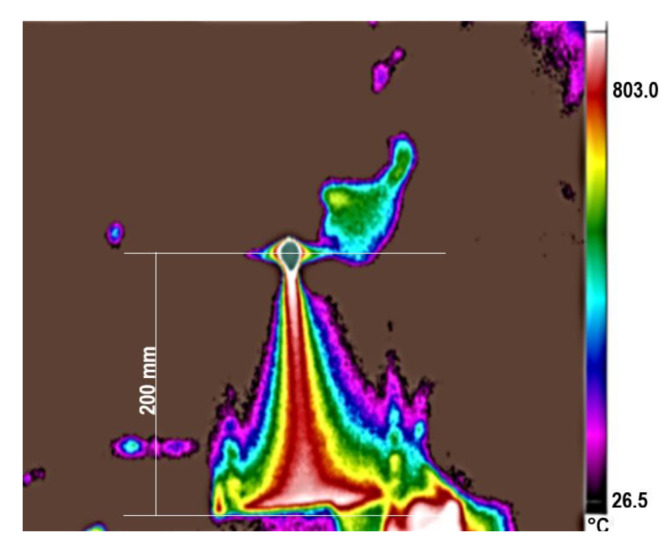
Infrared thermal image from the electric arc spray process.

**Figure 8 materials-16-04182-f008:**
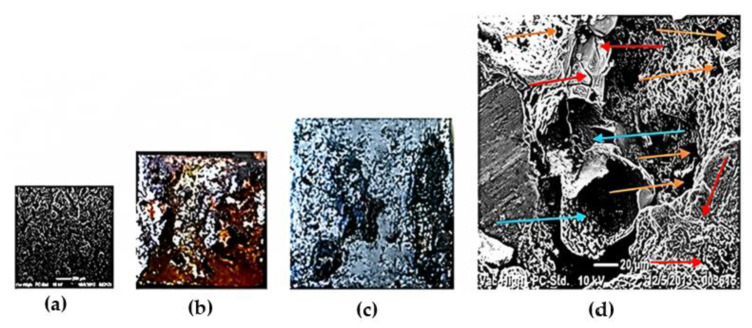
Morphology of the 140MXC-530AS mixtures exposed to saline chamber: (**a**) SEM initial surface condition before testing; (**b**) Rust formation after saline chamber test; (**c**) Rust removal with ultrasonic cleaning; (**d**) SEM final surface condition after testing.

**Figure 9 materials-16-04182-f009:**
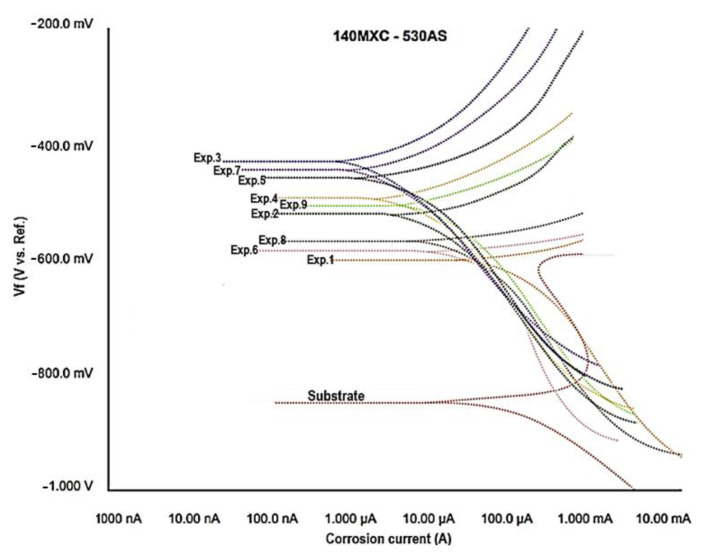
TAFEL test of the 140MXC-530AS mixtures.

**Figure 10 materials-16-04182-f010:**
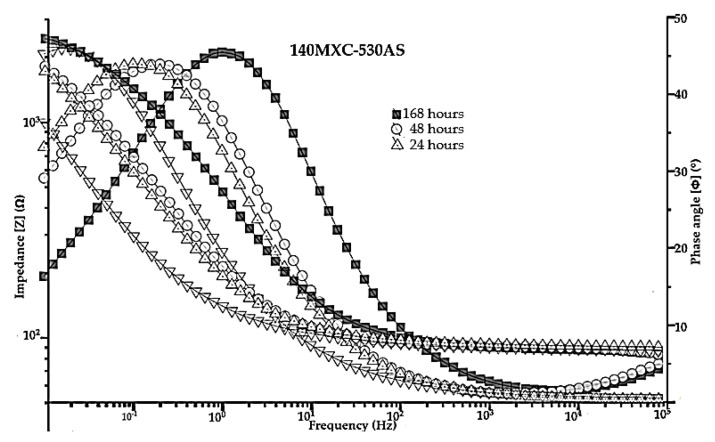
Bode diagrams of Experiment No 8 as a reference for the 140MXC-530AS mixture.

**Figure 11 materials-16-04182-f011:**
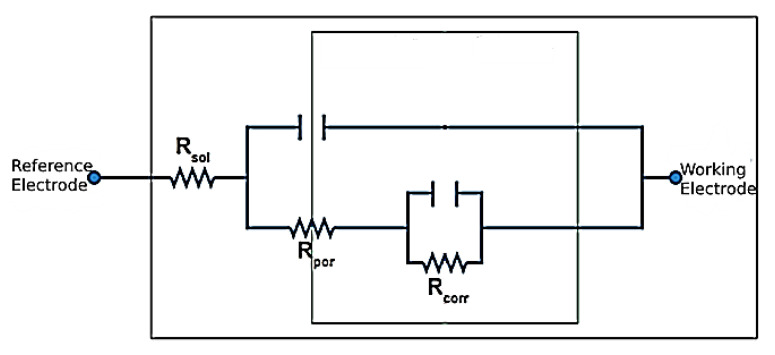
Equivalent circuit model considered for the EIS test in the 140MXC-530AS mixtures. Source: Adapted from [48].

**Figure 12 materials-16-04182-f012:**
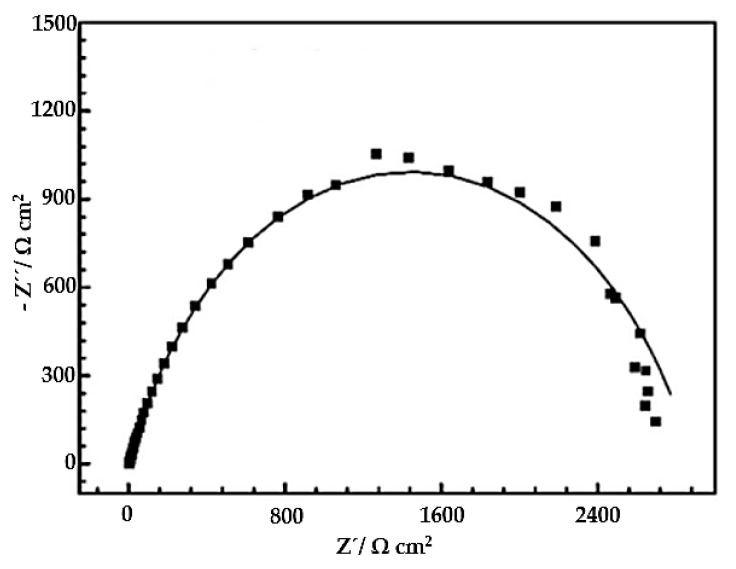
Nyquist diagram adapted for equivalent circuit model in EIS test of the 140MXC-530AS mixtures. Source: Adapted from [48].

**Figure 13 materials-16-04182-f013:**
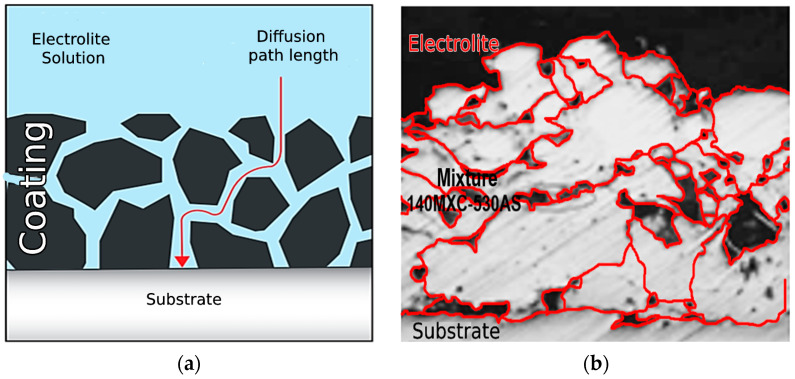
Corrosion mechanism present in the 140MXC-530AS mixtures (**a**) Source: Adapted from [48] (**b**) Experiment No 1.

**Table 1 materials-16-04182-t001:** Intensities relative energy (keV) for Lα and Kα levels of the elements present in the base coatings.

Element/Level	B	C	O	Si	Nb	Mo	Cr	Mn	Fe	Cu
**L(α) (KeV ±)**	/	/	/	/	2.1659	2.2932	0.5729	0.6374	0.7048	0.9297
**K(α) (KeV ±)**	0.1834	0.2774	0.5268	1.7398	16.5840	17.4446	5.4117	5.8951	6.3996	8.0413

**Table 2 materials-16-04182-t002:** Summary for the potentials and corrosion currents of the substrate and of the mixtures of the coatings.

Experiment	Substrate	1	2	3	4	5	6	7	8	9
**Icorr (µA ±)**	15.1	24.9	5.7	0.8	3.2	2.1	9.9	0.9	8.1	6.2
**Ecorr (mV ±)**	850	600	525	425	490	455	585	445	575	505

## Data Availability

Data can be requested to the authors.

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
