# Peer review of "Effect of the Surface Chemical Composition on the Corrosion Resistance in the Mixture of FeCrMoNbB (140MXC) and FeCMnSi (530AS) Coatings Produced with the Electric Wire Arc Spraying Technique: Part I"

_materials, 2023, doi:10.3390/ma16114182_

Round 1
Reviewer 1 Report
Please check the attached file.

Author Response
"Please see the attachment."

Reviewer 2 Report
Comment 1: The keyword "dissimilar" is not the core word of this article, so it is recommended to delete it.
Comment 2: According to the requirements of the journal, the first letter of the keyword should be capitalized.
Comment 3: It is recommended to add a paragraph at the end of the Introduction to introduce the purpose of the research topic and summarize the main content of the paper.
Comment 4: Line 69-75: The equipment used in the electric wire arc spraying technique is introduced on page 2 of this paper, and it is recommended to add a schematic diagram of this equipment.
Comment 5: The pictures such as XPS, EDX, XRD, etc. in the paper are blurry and need to be replaced with clear pictures.
Comment 6: In Table 2, incorrect figures appear and need to be modified.
Comment 7: Table 1 and Table 3 are duplicated, and it is suggested to delete one.
Comment 8: Line 323: It is recommended to add the impedance phase angle diagram of the high frequency band to the Figure7 Bode plot to facilitate a more intuitive reaction of electrolyte impedance, so as to correspond to the fitted equivalent circuit.
Comment 9: Figure 8 shows an equivalently fitted circuit, but lacks fitted data and an interpretation of the obtained curve, which needs to be added.
Comment 10: For journal papers published in the past 5 years are less cited, it is recommended to add several journal papers in recent years.
Reviewer 3 Report
The abstract does not provide information about the method of creating materials and the results obtained.
Also, in the introduction, when analyzing the literature, either the equipment used in these works is listed, but specific results and conclusions from the experiments are not given (lines 50-59), or vice versa, the positive effect of the coating is indicated, but it is not described exactly how it was obtained (lines 46-49). The rest of the section is made up of general words that do not introduce the reader to the essence of the problem under study. Description of the purpose and objectives of the study is completely absent.
On line 117, the sentence probably begins with a mistake.
Table 2 has a bad format.
Figure 1 crawls out of the margins.
Figure 5 - it is impossible to understand the scale of the images, and it is not clear why the drawings were arranged in this way.
Authors should be careful with degrees, the reader is not obliged to guess that, for example, ± 0.75x10-6 (line 382) is ± 0.75x0.000001, the text must be formatted.
In the chapter “results and discussion” there are a lot of specific results, but their discussion is shown little or not clearly, while each result should ultimately lead the authors to some kind of conclusion.
The conclusion looks unnecessarily cumbersome, it is not clear what conclusions the authors of the article come to as a result of their research.
Round 2
Reviewer 2 Report
I think this version meets the requirement of publication after detailed modification. Congratulation!
